# IFN-Gamma Expression in the Tumor Microenvironment and CD8-Positive Tumor-Infiltrating Lymphocytes as Prognostic Markers in Urothelial Cancer Patients Receiving Pembrolizumab

**DOI:** 10.3390/cancers14020263

**Published:** 2022-01-06

**Authors:** Toru Sakatani, Yuki Kita, Masakazu Fujimoto, Takeshi Sano, Akihiro Hamada, Kenji Nakamura, Hideaki Takada, Takayuki Goto, Atsuro Sawada, Shusuke Akamatsu, Takashi Kobayashi

**Affiliations:** 1Department of Urology, Graduate School of Medicine, Kyoto University, Kyoto 606-8507, Japan; sktntoru5@kuhp.kyoto-u.ac.jp (T.S.); kitayuki@kuhp.kyoto-u.ac.jp (Y.K.); sanotake@kuhp.kyoto-u.ac.jp (T.S.); ahamada@kuhp.kyoto-u.ac.jp (A.H.); kenji924@kuhp.kyoto-u.ac.jp (K.N.); tama35@kuhp.kyoto-u.ac.jp (H.T.); goto@kuhp.kyoto-u.ac.jp (T.G.); atsuro7@kuhp.kyoto-u.ac.jp (A.S.); akamats@kuhp.kyoto-u.ac.jp (S.A.); 2Department of Diagnostic Pathology, Graduate School of Medicine, Kyoto University, Kyoto 606-8507, Japan; fujimasa@kuhp.kyoto-u.ac.jp

**Keywords:** urothelial carcinoma, pembrolizumab, immune checkpoint inhibitor, interferon-gamma, CD8

## Abstract

**Simple Summary:**

Pembrolizumab, an immune checkpoint inhibitor, has shown therapeutic benefit for advanced urothelial carcinoma (aUC) patients, but only a limited population achieves a long-term response. Prediction of treatment outcomes for aUC patients receiving immune checkpoint inhibitors is a clinical challenge. We assessed the associations between the expression of multiple immune markers in the tumor microenvironment using immunohistochemistry of tumor tissues obtained from 26 aUC patients who received second-line pembrolizumab treatment. We found that high infiltration of CD8-positive lymphocytes was significantly associated with a favorable objective response and overall and progression-free survival. Furthermore, expression of interferon-gamma (IFNγ) showed a significant positive correlation with post-progression survival. Finally, we demonstrated that the coincidence of low infiltration of CD8-positive lymphocytes and low IFNγ expression was an independent prognostic factor for an unfavorable response to pembrolizumab.

**Abstract:**

Although immune checkpoint inhibitors have shown benefit for advanced urothelial carcinoma (aUC) patients, prognostication of treatment efficacy and response duration remains a clinical challenge. We evaluated the expression of immune markers in the tumor microenvironment and assessed their associations with response to and survival after pembrolizumab treatment in 26 aUC patients. High levels of CD8^+^ tumor-infiltrating lymphocytes (TILs) were associated with favorable objective responses (23.0% vs. 15.3%, *p* = 0.0425), progression-free survival (median, 8.8 vs 2.1 months; hazard ratio (HR), 0.24; 95% confidence interval (CI), 0.07–0.66, *p* = 0.0060), and overall survival (median, >24.0 vs. 5.3 months; HR, 0.17; 95% CI, 0.04–0.56, *p* = 0.0034) compared with low levels. High interferon-gamma (IFNγ) expression levels were associated with longer post-progression survival (median, 4.9 vs. 1.0 months; HR, 0.18; 95% CI, 0.04–0.59, *p* = 0.0027) compared with low expression. Multivariate analysis incorporating clinical prognosticators demonstrated that the coincidence of low CD8^+^ T cells/IFNγ was an independent factor for unfavorable overall survival after pembrolizumab treatment (HR, 4.07; 95% CI, 1.36–12.73; *p* = 0.0125). The combination of low CD8^+^ TILs and IFNγ expression was an independent prognostic factor with predictive ability equivalent to previously reported clinical prognosticators.

## 1. Introduction

Advanced urothelial carcinoma (aUC) that is locally advanced, unresectable, or metastatic has a poor prognosis [1]. Cisplatin-based chemotherapy has been used mainly as the first-line therapy for aUC over the past several decades with an overall survival (OS) rate of 9–15 months. However, in aUC patients with resistance to platinum therapy, the median OS is only 5–7 months [2,3]. In addition, approximately 50% of aUC patients are ineligible for cisplatin therapy [4], and there are few treatment options for chemo-resistant or cisplatin-ineligible aUC patients.

Over the last several years, results of multiple clinical trials have led the U.S. Food and Drug Administration to approve five immune checkpoint inhibitors (ICIs) for aUC [5,6,7]. The appearance of novel immunotherapies has changed the treatment landscape substantially for aUC. However, the population who can benefit from a prolonged OS after ICI treatment is limited. For example, in the phase 3 KEYNOTE-045 trial using second-line pembrolizumab, the >2-year follow-up period showed an OS of 44.2% and an objective response rate of 21.2% at 24 months [8,9]. Furthermore, pembrolizumab treatment resulted in serious adverse events in 12% of patients, such as colitis, pneumonitis, and interstitial lung disease. Additionally, 6.8% of patients receiving pembrolizumab discontinued the study treatment because of these treatment-related adverse events. Thus, there is an urgent unmet need to develop methods for predicting the treatment efficacy and safety of ICIs for aUC patients. 

One of the most frequently reported potential biomarkers in tumor tissues is programmed cell death-ligand 1 (PD-L1). Multiple studies have investigated the role of PD-L1 protein expression as a predictive marker for the efficacy of ICIs [7]. These previous studies were consistent in reporting an association between high expression of PD-L1 and a favorable response to ICIs. However, the protocols for staining, evaluation, and reporting PD-L1 expression have not been standardized, and highly diverse protocols, antibodies, and cutoff values have been reported. Additionally, concerns have arisen regarding the predictive ability of PD-L1 expression for ICI treatment responses [9]. Indeed, the clinical use of PD-L1 tumor expression as a companion diagnostic marker for efficacy of ICIs is limited for first-line ICI application in cisplatin-ineligible patients [5,6].

Tumor-infiltrating lymphocytes (TILs), particularly CD8^+^ T cells, are considered to be key effectors of the anti-tumor immune activities of ICIs. Several studies have reported a correlation between tumor-infiltrating CD8^+^ T cells and a favorable response to ICIs [10,11]. CD8^+^ T cells within the tumor microenvironment (TME) interact with various other immune cells, including helper T cells, regulatory T cells, dendritic cells, and macrophages [12]. 

Recently, interferon-gamma (IFNγ) has attracted increasing attention as a key factor that modulates the TME [13]. IFNγ was first identified as a cytokine that evoked an immune response to viral infection [14] and played an important role in cancer immunity. IFNγ contributes to recruitment of T lymphocytes, activates TILs by promoting dendritic cell-mediated antigen presentation, and directly induces apoptosis of cancer cells [15,16,17]. Several studies have reported that treatment with ICIs increased the production of IFNγ in the TME, which enhanced tumor regression [18,19]. Another study reported that defects in IFNγ signaling conferred ICI resistance [20]. These findings imply that the anti-tumor effect of ICIs may be mediated by IFNγ secretion in the TME. However, the clinical significance of IFNγ expression in the TME and its use as a predictive biomarker for ICI treatment response in aUC patients have not been well described. In this study, we evaluated multiple immune markers in the TME, including CD8^+^ T cells and IFNγ, and determined correlations between expression levels and treatment outcomes in aUC patients who received second-line pembrolizumab.

## 2. Materials and Methods

### 2.1. Cases and Specimens

We included 26 urothelial carcinoma patients with metastatic or unresectable tumors who had progressed after platinum-based chemotherapy and received subsequent pembrolizumab treatment from 2018 to 2020 at Kyoto University Hospital. The study was approved by the Institutional Review Board at Kyoto University Graduate School of Medicine (#G0052-17). Disease stage was determined according to the staging system from the 8th Union for International Cancer Control. Treatment effects were assessed according to the Response Evaluation Criteria in Solid Tumors (RECIST), ver. 1.1. Treatment effects were determined clinically in some cases where the RECIST criteria could not be applied.

### 2.2. Immunohistochemical (IHC) Staining

Formalin-fixed, paraffin-embedded histological sections were prepared from primary tumors obtained by transurethral resection of the bladder, total cystectomy, or nephroureterectomy. These surgeries were performed prior to systemic therapy from 2014 to 2020 at Kyoto University Hospital. IHC staining was performed as described previously [21]. Briefly, after deparaffinization and antigen retrieval, endogenous peroxidase activity was quenched using 0.3% H_2_O_2_ in methyl alcohol for 30 min. The glass slides were washed six times in phosphate-buffered saline (PBS) and blocked with 1% normal serum in PBS for 30 min. Subsequently, primary antibodies were applied overnight at 4 °C. The slides were then incubated for 40 min with biotinylated secondary antibody diluted at 1:300 in PBS. Avidin–biotin–peroxidase complex (ABC-Elite, Vector Laboratories) diluted at 1:100 in bovine serum albumin was applied for 50 min. After washing in PBS, the color reaction was initiated with 3,3′-diaminobenzidine tetrahydrochloride, and the nuclei were counterstained with hematoxylin.

CD8, CD4, and IFNγ were stained to confirm the immune response of the host patient in the primary tumor, and PD-L1 and TIGIT were selected to confirm the expression status of immune checkpoint molecules in the tumor microenvironment. The primary antibodies were as follows: anti-CD8 (M7103; DAKO; Carpenteria, CA, USA; 1:100), anti-CD4 (NCL-CD4-1F6; Leica Biosystems; Wetzlar, Germany; 1:100), anti-PD-L1 (ab228462; Abcam; Cambridge, UK; 1:100), anti-T cell immunoreceptor with Ig and ITIM domains (TIGIT) (ab243903; Abcam, 1:100), and anti-IFNγ (ab9657; Abcam, 1:400). Stained slides were reviewed and scored by two independent investigators under the supervision of a pathologist. To quantify antibody-positive cells, the numbers of stained cells were counted using three high-power fields per slide (×400 magnification). 

Our quantitative evaluation protocol was as follows: CD8^+^, CD4^+^, and TIGIT^+^ TIL densities were defined as the percentage of the total number of immune-shaped cells present within the tumor. A density of CD8^+^ and CD4^+^ TILs at <20% was considered to be low, whereas ≥20% was considered high infiltration, as previously reported [22]. For TIGIT^+^ TILs, a density < the median was considered low infiltration, and a density ≥ the median was considered a high infiltrate. 

The level of expression of PD-L1 in tumor cells (TC) and infiltrating immune cells (IC) was stratified to four groups based on the scores of 0, 1, 2, and 3, which were defined as previously described—score 0 ≤ 1%, 1% < score 1 ≤ 5%, 5% < score 2 ≤ 30%, and 30% < score 3 [23]. We divided the TCs and ICs into two groups each—a low expression group with a score of 0 or 1 and a high expression group with a score of 2 or 3. 

IFNγ expression in the tumor tissue was evaluated based on a combination of both the percentage and intensity of positively stained cells to generate H-scores according to a previous report [24]. A total score consisted of 3× the percentage of strongly stained cells, 2× the percentage of moderately stained cells, and 1× the percentage of weakly stained cells in the TME. A final score had a range of 0 to 300 [25]. Subsequently, an H-score < the median was low expression and an H-score ≥ the median was considered to be high expression.

### 2.3. Statistical Analysis

Associations between categorical variables were assessed with Fisher’s exact test. Spearman’s rank correlation coefficients were used to test associations between IFNγ expression and the other immunological biomarkers. OS was defined as the interval from the date of the initiation of pembrolizumab treatment to death or last contact. Progression-free survival (PFS) was defined as the interval from the date of the initiation of pembrolizumab to the date of disease progression. Post-progression survival (PPS) was defined as the interval from the date of disease progression during or after pembrolizumab treatment to death or last contact. Survival was analyzed using the Kaplan–Meier method and Cox regression analysis. The correlations among OS, PFS, and PPS were assessed by Spearman’s rank correlation coefficients and linear regression analyses without censored cases as described previously [26]. The correlations for survival were also tested using data from 450 metastatic urothelial carcinoma cases treated with pembrolizumab in another study cohort [27]. Some parameters that had clinical importance for immunotherapy at the time of pembrolizumab treatment were evaluated by multivariate tests to compare the predictive capacity of candidate immune biomarkers found in this study [27,28]. All statistical analyses were performed using JMP^®^ Pro 14.0.0 (SAS Institute; Cary, NC, USA) and R PredictABEL package, R version 4.1.2. *p* values < 0.05 were considered statistically significant.

## 3. Results

### 3.1. Baseline Characteristics

Baseline characteristics of the 26 study patients are shown in Table 1. The median (range) age at the initiation of pembrolizumab treatment was 72.5 (47–87) years. The median follow-up period from the initiation of pembrolizumab treatment to the date of the last follow-up was 6.4 (1.0–24.0) months. The median number of administrations of pembrolizumab was 5.5 cycles (1–25). The median OS of the 26 study patients was 7.0 (1.0–24.0) months, while PFS was 3.5 (0.1–24.0) months, and PPS was 2.4 (0.1–8.0) months (Figure 1C–E). As shown in Figure 1A,B, there were wide individual variations in PFS and PPS. Notably, in Figure 1B, high expression of IFNγ was related to longer PPS. The objective response rates were 0% for complete response, 38.5% for partial response, and 61.5% for stable or progressive disease (PD) (Table 2). 

### 3.2. Correlations among Immune Biomarkers

Representative photomicrographic images of IHC are shown in Figure 2. Correlation analyses of multiple immune biomarkers revealed that the expression of IFNγ was inversely correlated with that of TIGIT (Spearman’s rho = −0.4218, *p* = 0.0318, Table 3). 

### 3.3. Correlations among Survival Outcomes

PFS showed a significant and strong correlation with OS (Spearman’s rho = 0.766, *p* = 0.0001, R^2^ = 0.56, Figure 3A), while PPS showed a significant but mild correlation with OS (Spearman’s rho = 0.493, *p* = 0.0318, R^2^ = 0.42, Figure 3B). These data indicated that both PFS and PPS had significant impact on OS of aUC patients receiving pembrolizumab therapy. No significant correlation was observed between PFS and PPS (Spearman’s rho = −0.0663, *p* = 0.9463, R^2^ = 0.00028, Figure 3C). Similar correlations were observed in the 450-patient cohort described above [27] (PFS with OS: Spearman’s rho = 0.7299, *p* < 0.0001, R^2^ = 0.49, Figure 3D; PPS with OS: Spearman’s rho = 0.7022, *p* < 0.0001, R^2^ = 0.49, Figure 3E; PFS with PPS: Spearman’s rho = 0.2033, *p* = 0.8977, R^2^ = 0.00004, Figure 3F). Multiple linear regression analyses showed that both PFS (*p* < 0.0001) and PPS (*p* < 0.0001) were independently correlated with OS.

### 3.4. Correlations between the Expression of Immune Biomarkers and Survival Outcomes

A high level of CD8^+^ TILs was significantly correlated with longer PFS (median, 8.8 vs. 2.1 months; hazard ratio (HR), 0.24; 95% confidence interval (CI), 0.07–0.66; *p* = 0.0060) and OS (median, >24.0 vs. 5.3 months; HR, 0.17; 95% CI, 0.04–0.56; *p* = 0.0034) compared with low CD8^+^ TIL levels, while high IFNγ expression was significantly associated with longer PPS compared with low expression levels (median, 4.9 vs. 1.0 months; HR, 0.18; 95% CI, 0.04–0.59; *p* = 0.0027) (Figure 4A–F and Table 4). A high level of CD8^+^ TILs was also significantly associated with a favorable objective response rate compared with low levels (23.0% vs. 15.3%, *p* = 0.0425, Table 2). PD-L1 expression in TCs or ICs was not associated with survival outcome after pembrolizumab treatment.

### 3.5. Predictive Capacity of Concurrently Low CD8^+^ T Cell Infiltration and IFNγ Expression for Survival Outcome in aUC Patients Treated with Pembrolizumab as a Second-Line Therapy

Multivariate Cox proportional hazard analyses incorporated clinical prognostic factors, performance status ≥2, the presence of liver metastasis, and neutrophil-to-lymphocyte ratios (NLR) at the time of pembrolizumab treatment [27,28]. These analyses revealed that the coincidence of low CD8^+^ T cells and IFNγ expression (HR, 4.07; 95% CI, 1.36–12.73; *p* = 0.0125) as well as high NLR (HR, 14.45; 95% CI, 2.12–108.06; *p* = 0.0067) were independent factors for unfavorable OS after pembrolizumab treatment. In PFS, only the NLR was an independent factor for an unfavorable outcome (HR, 26.451; 95% CI, 3.76–202.62; *p* = 0.0011). For PPS, concurrently low CD8^+^ T cells and IFNγ expression (HR, 7.20; 95% CI, 2.13–27.88; *p* = 0.0014) and the presence of liver metastasis (HR, 4.84; 95% CI, 1.29–18.16; *p* = 0.020) were independent factors for unfavorable outcomes (Table 5).

Kaplan–Meier curves revealed that the coincidence of low CD8^+^ T cells and low IFNγ was significantly associated with shorter PPS (median, 0.78 vs. 4.96 months; HR, 5.94; 95% CI, 1.91–20.23; *p* < 0.001) compared with high levels. However, the association with shorter OS did not reach statistical significance (median, 5.08 vs. 10.13 months; HR, 2.49; 95% CI, 0.93–6.47; *p* = 0.0517) (Figure 4G–I).

Although NRI and IDI analyses did not show statistically significant difference between IFNγ alone and in combination with CD8, the potential of the combination as a prognosticator should be further studied in the future large-scale studies.

## 4. Discussion

We determined the correlations between IFNγ expression, infiltration of immune cells, and expression of checkpoint molecules in the TME of primary tumors from aUC patients. We demonstrated that high levels of CD8^+^ TILs were significantly associated with a favorable objective response and longer OS, while high IFNγ expression in the tumor was significantly correlated with longer PPS. Furthermore, coincidence of low CD8 infiltration and low IFNγ secretion was associated with an unfavorable prognosis. These data suggest that the combinatorial evaluation of CD8^+^ TILs and IFNγ expression may provide a novel prognostic biomarker for patients receiving pembrolizumab treatment for chemorefractory aUC. 

Tumor infiltration by effector T cells is considered to be mandatory for effective treatment with ICIs [29,30]. Indeed, recent studies have demonstrated that a high level of infiltrating CD8^+^ T cells was associated with favorable radiological response and oncological outcomes after ICI treatment [10,11,31], which is consistent with our results. 

Not surprisingly, PFS was strongly correlated with OS; additionally, PPS was another factor that correlated with OS in this study using two independent cohorts. Although the significance of PPS in aUC after pembrolizumab treatment is not well understood, the significant correlations between PPS and OS in the two independent cohorts analyzed in the present study strongly suggest the potential importance of PPS in addition to PFS. In the phase 3 OAK trial, a post-hoc analysis of atezolizumab versus docetaxel in advanced non-small-cell lung cancer (NSCLC) demonstrated that PPS in the atezolizumab group varied widely [32]. Specifically, PPS was 12.7 months (95% CI, 9.3–14.9) in patients continuing atezolizumab treatment, 8.8 months (95% CI, 6.0–12.1) in those switching to other anticancer therapy, and 2.2 months (95% CI, 1.9–3.4) for those receiving no further therapy. These results suggested that continuation of anti-PD-1/PD-L1 treatment until loss of clinical benefit contributed to favorable PPS. Another possible explanation was better clinical status at the time of PD, allowing for atezolizumab treatment beyond PD or subsequent anticancer treatment. Additionally, in our cohort, PPS was 4.97 months (95% CI, 1.60–8.07) in patients with persistent pembrolizumab as long as possible after PD (*n* = 7) and PPS was 1.2 months (95% CI, 0.4–2.7) in patients with no treatment after PD (*n* = 15). Continuation of pembrolizumab tended to correlate with favorable PPS (*p* = 0.053) although the difference did not reach a statistical significance. This could have been a β error due to the small sample size and should be further validated in a future study.

Factors that may affect PPS in patients with advanced cancer receiving anti-PD-1/PD-L1 treatment have not been fully elucidated. Our results suggest that the prediction of PPS may be possible by using a biomarker, such as IFNγ expression, and this predictive potential would be useful for clinical decision-making and/or patient counseling in aUC management. Further investigations will be required to understand the biological and clinical determinants of PPS in aUC patients receiving pembrolizumab.

The interaction between IFNγ and CD8^+^ T cells in the TME plays an important role in anti-tumor immune responses induced by ICIs. In the CheckMate-275 phase 2 clinical trial, 270 patients with platinum-resistant metastatic UC were treated with the anti-PD-1 antibody nivolumab; the presence of higher tumor infiltration by CD8^+^ T cells prior to treatment was associated with favorable PFS and OS after nivolumab treatment [33]. Furthermore, enrichment of a 25-gene IFNγ signature was associated with the response to nivolumab in aUC patients after the platinum-based chemotherapy. Another study reported RNA sequencing results for tumor tissues obtained from 62 locally advanced or metastatic UC patients and 97 advanced NSCLC patients. This study demonstrated that a 4-gene IFNγ signature was associated with a favorable response to durvalumab treatment in both types of cancer [34]. Additionally, IFNγ has been reported to enhance the expression of major histocompatibility class 1 molecules and promote the presentation of tumor-specific antigens [35]. Enhancement of anti-tumor immunity by upregulating antigen recognition and tumor infiltration by T cells then stimulates CD8^+^ T cells to secrete IFNγ, which forms a positive feedback loop [36]. 

TIGIT is a co-inhibitory receptor mainly expressed on the surface of CD8^+^ T cells, NK cells, and regulatory T cells [37,38]. TIGIT has been reported to contribute to immune evasion and to be correlated with poor prognosis in several cancers [39,40]. This study showed that the expression of TIGIT in TILs was inversely correlated with the expression of IFNγ in the TME. Our results were consistent with a recent study demonstrating that TIGIT expression was inversely correlated with the expression of cytotoxic cytokines, including IFNγ, from CD8^+^ T cells infiltrating colorectal cancer [41]. The previous study also showed that other co-inhibitory receptors, such as PD-1, T cell immunoglobulin and mucin domain-containing protein 3, and lymphocyte activating gene-3 were expressed on the surface of TIGIT^+^ CD8^+^ T cells.

In this study, PD-L1 expression in TCs or ICs was not associated with favorable clinical outcomes after second-line pembrolizumab treatment. This result is in contrast to several clinical trials that showed a correlation between PD-L1 expression and prolonged OS [42] or benefit of anti-PD-1/PD-L1 treatment [9]. The contradictory results may be explained by the use of different antibodies and cutoff values for PD-L1 positivity. In the present study, we used the Abcam SP142 assay for PD-L1 IHC, which is different from the Dako 22C3 antibody used in the KEYNOTE-045 and KEYNOTE-052 studies [9,42].

There were several limitations in this study that should be acknowledged in addition to the retrospective design and small sample size. One limitation was that PPS was not tested in a control group that did not receive pembrolizumab, such as a group of patients treated with chemotherapy alone or not treated. Additionally, no patient received subsequent systemic therapy after pembrolizumab in this study. It is desirable to evaluate the correlation between the profile of additional therapy and the expression patterns of biomarkers to better understand the clinical roles of PPS. Furthermore, a cutoff value for IFNγ was not externally validated. A gene-expression signature was not evaluated, and, therefore, it is unclear whether high IFNγ expression represents an enriched IFNγ gene signature. Nonetheless, we believe that the present study demonstrates a novel combinatorial prognostic factor using CD8^+^ TIL levels and IFNγ expression and provides new insight into the mechanism of resistance of aUC to ICIs. Our results warrant further investigations including consistency of our findings with other ICIs and validation using gene-expression profiling data in larger-scale studies. 

## 5. Conclusions

High tumor infiltration by CD8^+^ lymphocytes was associated with longer OS and PFS of patients with aUC treated with pembrolizumab, and high expression of IFNγ in the TME was significantly correlated with favorable PPS. The coincidence of low expression of CD8^+^ TILs and IFNγ was an independent prognostic factor that yielded a predictive ability equivalent to previously reported clinical prognostic factors.

## Figures and Tables

**Figure 1 cancers-14-00263-f001:**
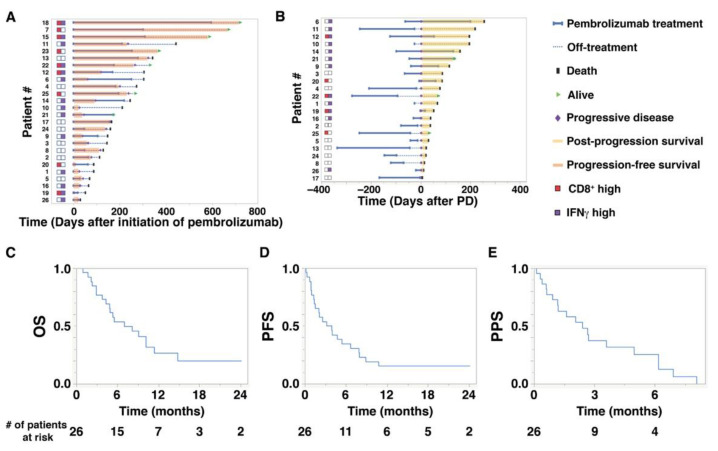
Overall survival (OS), progression-free survival (PFS), and post-progression survival (PPS) of aUC patients treated with pembrolizumab. (**A**) A swimmer plot illustrates OS and PFS (orange bars) of 26 patients who received pembrolizumab treatment. The status of CD8^+^ T cells (red squares) and interferon gamma (IFNγ) expression (purple squares) is indicated on the left. (**B**) A swimmer plot illustrates PPS (yellow bars) of 22 patients who eventually developed progressive disease (PD) during or after pembrolizumab treatment. The horizontal axis shows the number of days after the initiation of PD. (**C**–**E**) Kaplan–Meier plots for (**C**) OS, (D) PFS, and (**E**) PPS of 26 patients who received pembrolizumab treatment.

**Figure 2 cancers-14-00263-f002:**
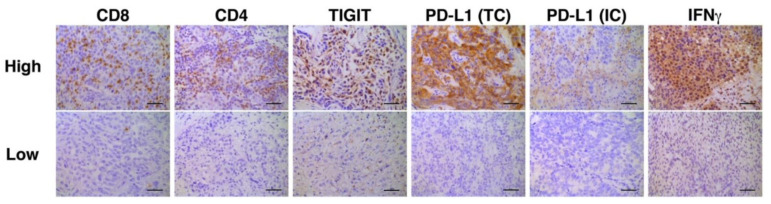
Representative immunohistochemistry stains for high and low expression of each biomarker. Bars represent 100 μm. PD-L1 staining in TC was defined as either partial or complete membranous staining of any intensity in tumor cells. PD-L1 staining in IC was defined as cytoplasmic or membranous staining of any intensity in immune cells. Only tumor-infiltrating ICs and the ICs which infiltrate surrounding the tumor area within the peritumoral stroma were included in IC scoring.

**Figure 3 cancers-14-00263-f003:**
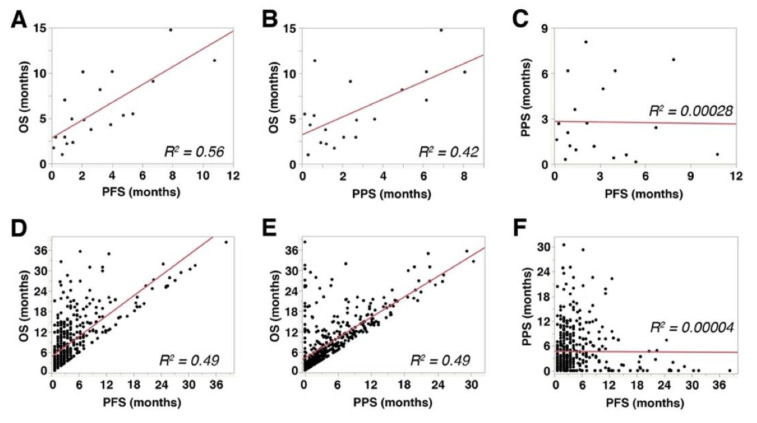
Correlations among overall survival (OS), progression-free survival (PFS), and post-progression survival (PPS): (**A**–**C**) The results from the cohort of 26 patients included in the present study. (**D**–**F**) The results from a separate cohort that comprised 450 metastatic urothelial carcinoma patients treated with pembrolizumab [27]. R^2^ represents linear regression.

**Figure 4 cancers-14-00263-f004:**
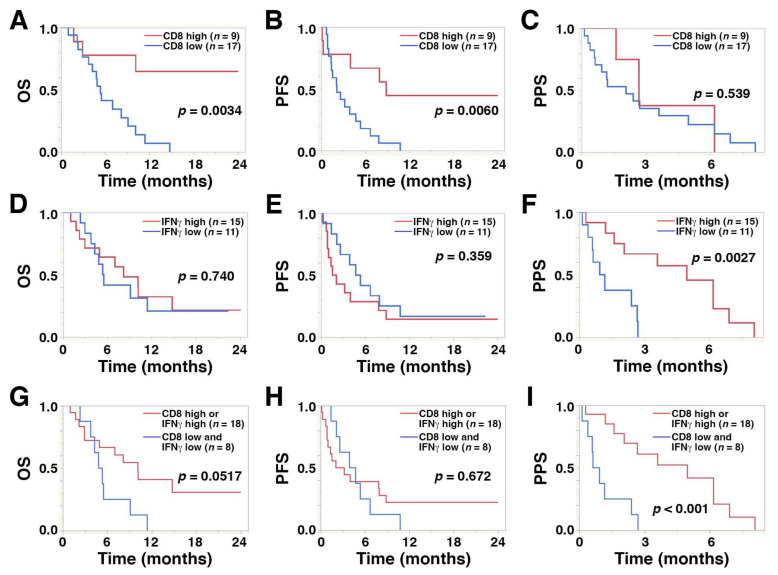
Kaplan–Meier plots displaying overall survival (OS), progression-free survival (PFS), and post-progression survival (PPS) for the 26 study patients with regard to high and low levels of (**A**–**C**) CD8^+^ T cells, (**D**–**F**) IFNγ, and (**G**–**I**) the combination of CD8^+^ T cells and IFNγ.

**Table 1 cancers-14-00263-t001:** Baseline characteristics of patients treated with second-line pembrolizumab for advanced urothelial carcinoma.

Patients, *n* = 26 (%)			
Age, years		Primary site	
Median (range)	72.5 (47–87)	Bladder cancer	18 (69.2)
Sex		Upper urinary tract	8 (30.7)
Female	5 (19.2)	Tumor status	
Male	21 (80.7)	pTa	0 (0)
Tobacco history		pTis	0 (0)
No	9 (34.6)	pT1	0 (0)
Yes	17 (65.3)	pT2	4 (15.3)
ECOG-PS		pT3	9 (34.6)
0	4 (15.3)	pT4	6 (23.0)
1	12 (46.1)	NA	7 (26.9)
2	5 (19.2)	Nodal status	
3	4 (15.3)	pN0	14 (53.8)
4	1 (3.8)	pN1	1 (3.8)
NLR		pN2	2 (7.6)
Median (range)	4.07 (1.03–28.84)	pN3	1 (3.8)
Prior chemotherapy		NA	8 (30.7)
GC	10 (38.4)	Metastatic site	
G-CBDCA	11 (42.3)	Liver	6 (23.0)
Both	4 (15.3)	Other	20 (76.9)
Others	1 (3.8)	No metastasis	1 (3.8)
Prior RT		Histology	
Yes	7 (26.9)	UC	21 (80.7)
No	19 (73.0)	Other	5 (19.2)
RT after Pembrolizumab		Histological grade	
Yes	7 (26.9)	High	26 (100.0)
No	19 (73.0)	Low	0 (0)

ECOG-PS, Eastern Cooperative Oncology Group-performance status; GC, gemcitabine, cisplatin; G-CBDCA, gemcitabine, carboplatin; NLR, neutrophil-to-lymphocyte ratio; pN, pathologic lymph node stage; pT, pathologic tumor stage; RT, radiation therapy; UC, urothelial carcinoma.

**Table 2 cancers-14-00263-t002:** Correlations between high and low biomarker expression and ORR.

		PR, *n* (%)	SD or PD, *n* (%)	*p* Value *
Patients		10 (38.5)	16 (61.5)	
IFNγ	High	4 (15.3)	10 (38.4)	-
	Low	6 (23.0)	6 (23.0)	0.9367
CD8	High	6 (23.0)	3 (11.5)	-
	Low	4 (15.3)	13 (50.0)	0.0425 ^†^
CD4	High	5 (19.2)	5 (19.2)	-
	Low	5 (19.2)	11 (42.3)	0.2929
PD-L1, TC	High	4 (15.3)	9 (34.6)	-
	Low	6 (23.0)	7 (26.9)	0.8869
PD-L1, IC	High	2 (7.6)	8 (30.7)	-
	Low	8 (30.7)	8 (30.7)	0.977
TIGIT	High	6 (23.0)	8 (30.7)	-
	Low	4 (15.3)	8 (30.7)	0.464

IC, infiltrating immune cell; IFNγ, interferon gamma; ORR, objective response rate; PD, progressive disease; PR, partial response; SD, stable disease; TC, tumor cell; TIGIT, T cell immunoreceptor with Ig and ITIM domains; * *p* values were obtained using Fisher’s exact test; ^†^ significant result, *p* < 0.05.

**Table 3 cancers-14-00263-t003:** Correlations between IFNγ expression and other immune biomarkers.

		Spearman’s rho	*p* Value *
IFNγ	CD8	0.0617	0.7646
	CD4	−0.0741	0.7189
	PD-L1, TC	0.063	0.7599
	PD-L1, IC	−0.0338	0.87
	TIGIT	−0.4218	0.0318 ^†^

IC, infiltrating immune cell; IFNγ, interferon gamma; PD-L1, programmed cell death-ligand 1; TC, tumor cell; TIGIT; T cell immunoreceptor with Ig and ITIM domains;. ** p* values were obtained using Fisher’s exact test; ^†^ significant result, *p* < 0.05

**Table 4 cancers-14-00263-t004:** Univariate Cox regression analysis of various biomarkers for OS, PFS, and PPS after pembrolizumab treatment.

		OS		PFS		PPS	
		HR (95% CI)	*p* *	HR (95% CI)	*p* *	HR (95% CI)	*p* *
**IFNγ**	High	Ref.	-	1.48 (0.63–3.52)	0.3642	Ref.	-
	Low	1.17 (0.46–2.91)	0.741	Ref.	-	5.37 (1.69–20.40)	0.0042 ^†^
**CD8**	High	Ref.	-	Ref.	-	Ref.	-
	Low	5.58 (1.78–24.65)	0.0021 ^†^	4.12 (1.51–13.34)	0.0047 ^†^	1.47 (0.47–6.39)	0.5338
**CD4**	High	Ref.	-	Ref.	-	Ref.	-
	Low	1.79 (0.70–5.10)	0.2272	1.46 (0.62–3.68)	0.3904	1.27 (0.48–3.72)	0.6371
**PD-L1, TC**	High	1.001 (0.40–2.49)	0.9976	1.16 (0.49–2.73)	0.7276	1.006 (0.38–2.69)	0.9901
	Low	Ref.	-	Ref.	-	Ref.	-
**PD-L1, IC**	High	1.26 (0.46–3.23)	0.6377	1.48 (0.60–3.47)	0.3819	Ref.	-
	Low	Ref.	-	Ref.	-	1.15 (0.44–3.17)	0.7797
**TIGIT**	High	Ref.	-	Ref.	-	2.06 (0.77–5.81)	0.1479
	Low	1.003 (0.40–2.50)	0.9943	1.38 (0.59–3.24)	0.4498	Ref.	-

CI, confidence interval; HR, hazard ratio; IC, infiltrating immune cell; IFNγ, interferon gamma; OS, overall survival; PD-L1, programmed cell death-ligand 1; PFS, progression-free survival; PPS, post-progression survival; Ref, reference; TC, tumor cell; TIGIT, T cell immunoreceptor with Ig and ITIM domains. * *p* values were obtained using the likelihood ratio test; ^†^ significant result, *p* < 0.05.

**Table 5 cancers-14-00263-t005:** Multivariate Cox regression analysis incorporating several clinical factors to evaluate the predictive capacity of the combination of low CD8^+^ T cells and IFNγ levels.

		OS		PFS		PPS	
		HR (95% CI)	*p* *	HR (95% CI)	*p* *	HR (95% CI)	*p* *
Low CD8^+^ T cells and low IFNγ	Yes	4.07 (1.36–12.74)	0.0125 ^†^	1.20 (0.46–2.97)	0.703	7.21 (2.14–27.89)	0.0014 ^†^
	No	Ref.	-	Ref.	-	Ref.	-
NLR		14.46 (2.13–108.07)	0.0067 ^†^	26.45 (3.76–202.62)	0.0011 ^†^	0.62 (0.09–4.10)	0.619
PS	≥2	1.305 (0.41–3.84)	0.635	1.22 (0.43–3.19)	0.695	3.44 (0.96–13.67)	0.058
	<2	Ref.	-	Ref.	-	Ref.	-
Liver metastasis	Yes	2.91 (0.83–9.49)	0.093	1.40 (0.43–3.87)	0.544	4.84 (1.29–18.16)	0.020 ^†^
	No	Ref.	-	Ref.	-	Ref.	-

CI, confidence interval; HR, hazard ratio; IFNγ, interferon gamma; NLR, neutrophil-to-lymphocyte ratio; OS, overall survival; PFS, progression-free survival; PPS, post-progression survival; PS, performance status; Ref, reference. * *p* values were obtained using the likelihood ratio test; ^†^ significant result, *p* < 0.05.

## Data Availability

The data underlying this article cannot be shared publicly to maintain the privacy of individuals that participated in the study. The data will be shared upon reasonable request to the corresponding author.

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
