# Peer review of "IFN-Gamma Expression in the Tumor Microenvironment and CD8-Positive Tumor-Infiltrating Lymphocytes as Prognostic Markers in Urothelial Cancer Patients Receiving Pembrolizumab"

_cancers, 2022, doi:10.3390/cancers14020263_

Round 1

Reviewer 1 Report

The manuscript is well written and the two most significant findings that being,  increased CD8+ lymphocytes effect on survival and objective responses and the effect of high IFN-gamma on post-progression survival, are well discussed and reviewed. The study is limited by its small size but these findings are intriguing and provide some basis for a larger confirmatory study.

The primary weaknesses are well discussed by the authors which including the small sample size and retrospective nature.  In addition, the post progression free survival is quite short and there is no discussion with regard to whether patients went on the receive additional therapy in the third and fourth line setting; which would clear impact PPS results especially if there was an imbalance in next line treatment between high and low expressions.  

The study did not find correlation with response and PD-L1 expression which does differ from other studies but agree this findings is inconsistent also in other studies; and has not been proven a reliable biomarker.

Minor comment is that the title only reflects the finding of high IFN-gamma on PPS and not the equally important finding with regard to CD8 positivity.

Author Response

General comment

The manuscript is well written and the two most significant findings that being, increased CD8+ lymphocytes effect on survival and objective responses and the effect of high IFN-gamma on post-progression survival, are well discussed and reviewed. The study is limited by its small size but these findings are intriguing and provide some basis for a larger confirmatory study.

Response: The authors appreciate the Reviewer for the positive comments. The authors have carefully addressed each specific comment from the Reviewer. Please see our point-by-point responses below.

Specific comments

Q1. The primary weaknesses are well discussed by the authors which including the small sample size and retrospective nature. In addition, the post progression free survival is quite short and there is no discussion with regard to whether patients went on the receive additional therapy in the third- and fourth-line setting; which would clear impact PPS results especially if there was an imbalance in next line treatment between high and low expressions. The study did not find correlation with response and PD-L1 expression which does differ from other studies but agree this findings is inconsistent also in other studies; and has not been proven a reliable biomarker. 

Response: The authors appreciate the Reviewer for the thoughtful comments on the important motivation and core concept of the present study. The authors totally agree with the Reviewer that subsequent anticancer therapies after disease progression against pembrolizumab treatment can affect PPS. In the present study, however, no patient received additional systemic therapy after pembrolizumab reflecting absence of approved treatment in the post-pembrolizumab setting in Japan. We have mentioned this issue in Discussion section of the revised manuscript as follows; “One limitation is that PPS was not tested in a control group that did not receive pembrolizumab, such as a group of patients treated with chemotherapy alone or not treated. Additionally, no patient received subsequent systemic therapy after pembrolizumab in this study. It is desirable to evaluate the correlation between the profile of additional therapy and the expression patterns of biomarker to better understanding about the clinical roles of PPS.

Q2. Minor comment is that the title only reflects the finding of high IFN-gamma on PPS and not the equally important finding with regard to CD8 positivity.

Response: The authors thank the reviewer for the critical comment. The authors totally agree with the Reviewer that the title should also reflect the role of CD8+ TILs as the predictive marker for ease of better understanding about the importance of this manuscript. The authors modified the title as follows; “IFN-gamma expression in the tumor microenvironment and CD8 positive tumor-infiltrating lymphocytes as prognostic markers in urothelial cancer patients receiving pembrolizumab”

Reviewer 2 Report

The authors presented an excellent work on the predictive value of immune infiltration detected by IHC in the aUC treated by Pembrolizumab. The article is well-written and give us transparent description of methods and interesting results supported by valid evidence. Still, some questions need clarification to improve the quality of this investigation.

Major comments:

  1. The cut-off value of IFNγ used in this study was the median. Did the authors ever test other cut-off values such as upper and lower quartile?
  2. In the discussion, the authors stated “Specifically, PPS was 12.7 months (95% CI, 9.3–14.9) in patients continuing atezolizumab treatment, 8.8 months (95% CI, 6.0–12.1) in those switching to other anticancer therapy, and 2.2 months (95% CI, 1.9–3.4) for those receiving no further therapy. These results suggested that continuation of anti-PD-1/PD-L1 treatment until loss of clinical benefit contributed to favorable PPS.” Did the authors perform statistical comparisons? The conclusion would be much of clinical value if supported by statistical evidence.
  3. The CD8 expression works as a good predictor of OS and PFS while IFNγ serves well as a PPS predictor. But it should be noticed that combination of these two biomarkers seems did not improve their predictive performance in oncological outcomes? It seems to me that the combination is not necessary. What is the performance of these two biomarkers as individual predictors in multi-variable regression? I strongly recommend a comparison between the combined model and individual biomarkers (eg. IDI, NRI analysis or DCA curves).

Minor comments:

  1. Some letters did not show well in the pdf. For example, the IFNγ in the footnotes of Tables. Please check the manuscript file thoroughly.
  2. The resolution and quality of Fig 1 need improvements for better presentation. Fig 1A and 1B should have similar size and they can not be easily read due to poor quality and small size. The fonts in the fig are also quite inconsistent. Please modify the fonts and the plots.
  3. The line between body and footnotes in the Table 3 is missing, please modify.

Author Response

General comment

The authors presented an excellent work on the predictive value of immune infiltration detected by IHC in the aUC treated by Pembrolizumab. The article is well-written and give us transparent description of methods and interesting results supported by valid evidence. Still, some questions need clarification to improve the quality of this investigation.

Response: The authors appreciate the Reviewer for the positive comments. The authors have carefully addressed each specific comment from the Reviewer. Please see our point-by-point responses below.

Specific comments

Q1. The cut-off value of IFNg used in this study was the median. Did the authors ever test other cut-off values such as upper and lower quartile?

Response: The authors thank the reviewer for the thoughtful comments. Median value as the cut-off was verified using receiver-operating characteristic analysis for concordance index and univariate cox proportional hazard model for best hazard ratio and p value. As for other cut-off values, the upper quartile yielded no significant prognostications for any oncological outcomes analysed in the present study, whereas the lower quartile yielded a similar prognosticative ability for unfavourable PPS to the median value. However, we selected the median value because of better distribution of the patients between the separated groups. 

Q2. In the discussion, the authors stated “Specifically, PPS was 12.7 months (95% CI, 9.3–14.9) in patients continuing atezolizumab treatment, 8.8 months (95% CI, 6.0–12.1) in those switching to other anticancer therapy, and 2.2 months (95% CI, 1.9–3.4) for those receiving no further therapy. These results suggested that continuation of anti-PD-1/PD-L1 treatment until loss of clinical benefit contributed to favorable PPS.” Did the authors perform statistical comparisons? The conclusion would be much of clinical value if supported by statistical evidence.

Response: The authors thank the Reviewer for the precious comments. In our cohort, PPS was 4.97 months (95% CI, 1.60–8.07) in patients with persistent Pembrolizumab as long as possible after PD (n = 7) and PPS was 1.2 months (95% CI, 0.4–2.7) in patients with no treatment after PD (n = 15). Continuation of pembrolizumab tended to correlate with favorable PPS (p = 0.053) although the difference did not reach a statistical significance. This can be a b error due to the small sample size and should be further validated in the future study. According to the helpful comments, some descriptions have been added to Discussion section of the revised manuscript as follows; “Another possible explanation was better clinical status at the time of PD, allowing for atezolizumab treatment beyond PD or subsequent anticancer treatment. Additionally, in our cohort, PPS was 4.97 months (95% CI, 1.60–8.07) in patients with persistent pembrolizumab as long as possible after PD (n = 7) and PPS was 1.2 months (95% CI, 0.4 -2.7) in patients with no treatment after PD (n = 15). Continuation of pembrolizumab tended to correlate with favorable PPS (p = 0.053) although the difference did not reach a statistical significance. This can be a b error due to the small sample size and should be further validated in the future study.

Q3. The CD8 expression works as a good predictor of OS and PFS while IFNg serves well as a PPS predictor. But it should be noticed that combination of these two biomarkers seems did not improve their predictive performance in oncological outcomes? It seems to me that the combination is not necessary. What is the performance of these two biomarkers as individual predictors in multi-variable regression? I strongly recommend a comparison between the combined model and individual biomarkers (eg. IDI, NRI analysis or DCA curves).

Response: The authors thank the reviewer for the critical comment. The authors performed NRI and IDI analyses for PPS to compare CD8 or IFNg alone with the combination of the two.

Although the combination did not show statistically significant difference, it demonstrated the possibility of better prognostication in PPS compared with IFNg alone. The authors consider that the potential of the combination as a prognosticator should be further studied in the future large-scale studies and decided to present in the present study. This issue is described in the end of the Result section as follows; “Although NRI and IDI analyses did not show statistically significant difference between IFNg alone and in combination with CD8, the potential of the combination as a prognosticator should be further studied in the future large-scale studies.

Q4. Some letters did not show well in the pdf. For example, the IFNg in the footnotes of Tables. Please check the manuscript file thoroughly.

Response: The authors thank the Reviewer for pointing out typos. All typos have been corrected in the revised manuscript accordingly.

Q5. The resolution and quality of Fig 1 need improvements for better presentation. Fig 1A and 1B should have similar size and they can not be easily read due to poor quality and small size. The fonts in the fig are also quite inconsistent. Please modify the fonts and the plots.

Response: The authors thank the Reviewer for the helpful comments. Figures 1A and 1B have been modified to improve the quality and readability.

Q6. The line between body and footnotes in the Table 3 is missing, please modify.

Response: The authors thank the Reviewer for the helpful comments. The table has been modified in the revised manuscript.

Reviewer 3 Report

Sakatani et al. present a study demonstrating the correlation of IFN-g in TME with post progression survival of UC patients after treated with Pembrolizumab. In this study, the authors used immunohistochemistry of tumors from 26 a UC patients, and the results showed low infiltration of CD8 positive lymphocytes and low IFN-g were independent prognostic factor for poor outcome. Overall, the study showed good data quality, and the manuscript is prepared in a proper format of Cancers. The comments for the authors are listed as follows:

  1. On page 6, line 192, please fix the typo of IFN@.
  2. On page 6, line 199, it is not knowing how these immune biomarkers were chosen for IHC staining? It would be nice to explain more. Meanwhile, please describe the meaning and how to distinguish TC and IC of PD-L1 staining in the legend of Figure 2 as well.
  3. Please use superscript to represent CD8 positive cells.
  4. There are different kinds of ICI approved for aUC patients. Is this finding specific to pembrolizumab or the phenomenon also observed in patients treated with other ICIs?

Author Response

General comment

Sakatani et al. present a study demonstrating the correlation of IFN-g in TME with post progression survival of UC patients after treated with Pembrolizumab. In this study, the authors used immunohistochemistry of tumors from 26 a UC patients, and the results showed low infiltration of CD8 positive lymphocytes and low IFN-g were independent prognostic factor for poor outcome. Overall, the study showed good data quality, and the manuscript is prepared in a proper format of Cancers.

Response: The authors appreciate the Reviewer for the positive comments. The authors have carefully addressed each specific comment from the Reviewer. Please see our point-by-point responses below.

Specific comments

Q1. On page 6, line 192, please fix the typo of IFNg.

Response: The authors thank the Reviewer for pointing out typos. All typos have been corrected in the revised manuscript accordingly.

Q2. On page 6, line 199, it is not knowing how these immune biomarkers were chosen for IHC staining? It would be nice to explain more. Meanwhile, please describe the meaning and how to distinguish TC and IC of PD-L1 staining in the legend of Figure 2 as well.

Response: The authors thank the Reviewer for the helpful comments. According to the precious comment, descriptions in the Materials and Methods section and legend for Figure 2 of the revised manuscript have been modified the of as follows;

Materials and Methods section: “CD8, CD4, and IFN were stained to confirm the immune response of the host patient in the primary tumor, and PD-L1 and TIGIT were selected to confirm the expression status of immune checkpoint molecules in the tumor microenvironment. The primary antibodies were as follows: …

Legend for Figure 2: “Representative immunohistochemistry stains for high and low expression of each biomarker. Bars represent 100 mm. PD-L1 staining in TC was defined as either partial or complete membranous staining of any intensity in tumor cells. PD-L1 staining in IC was defined as cytoplasmic or membranous staining of any intensity in immune cells. Only tumor-infiltrating ICs and the ICs which infiltrate surrounding the tumor area within the peritumoral stroma were included in IC scoring.

Q3. Please use superscript to represent CD8 positive cells.

Response: The authors thank the Reviewer for the helpful comments. According to the precious comment, the authors corrected all points in the revised manuscript.

Q4. There are different kinds of ICI approved for aUC patients. Is this finding specific to pembrolizumab or the phenomenon also observed in patients treated with other ICIs?

Response: The authors thank the reviewer for the critical comment. As the Reviewer pointed out, the interaction between CD8+ TILs and IFNg is also likely to play an important role in the antitumor immune response caused by other ICIs than pembrolizumab. In fact, there were some previous reports that patients with aUC with high CD8 TIL expression correlated with favorable PFS and OS of nivolumab [33], and that the expression of IFN signature correlated with favorable effects of durvalumab [34]. However, it is still not clear whether a low expression in both CD8+ TILs and IFNg is a prognosticator of the efficacy of other ICIs as well as pembrolizumab. Although the present study did not address the issue unfortunately, the authors have added some description to the Discussion section of the revised manuscript as follows; “Our results warrant further investigations including consistency of our findings with other ICIs and validation using gene expression profiling data in larger scale studies.

Round 2

Reviewer 2 Report

I would like to thank the authors' responses and revisions. The authors' response solved the questions and the revisions improved the quality of this article. I would recommend its publication in the present form.